# Antigen-agnostic microfluidics-based circulating tumor cell enrichment and downstream molecular characterization

**Evan N. Cohen** *, Gitanjali Jayachandran, Max R. Hardy, Ananya M. Venkata Subramanian, Xiangtian Meng, James M. Reuben**

Division of Pathology and Laboratory Medicine, Department of Hematopathology, The University of Texas MD Anderson Cancer Center, Houston, Texas, United States of America

* encohen@mdanderson.org

**Data Availability Statement:** All relevant data are within the manuscript and its Supporting Information files.

## Abstract

Circulating tumor cells (CTC) isolated from the peripheral blood of cancer patients by a minimally invasive procedure provide surrogate markers of the tumor that can be repeatedly sampled. However, the selection and enumeration of CTCs by traditional methods based on surface proteins like EPCAM may not detect CTCs with a mesenchymal phenotype. Here, we employed an antibody-agnostic platform, the Parsortix® PR1 system, which enriches CTCs based on cell size and membrane deformability. We evaluated the linearity, sensitivity, and specificity of the Parsortix PR1 system in tandem with 3 downstream molecular characterization techniques using healthy donor blood spiked with cultured cell lines. Signal amplification of mRNA using a QuantiGene 25-gene assay was able to quantitate multiple epithelial genes, including *CDH1*, *EGFR*, *ERBB2*, *KRT18*, and *MUC1*, from high numbers of spiked cells and was able to detect *KRT18* when only 50 MCF-7 or SUM190 cells were spiked into healthy donor blood. However, target amplification of mRNA by quantitative polymerase chain reaction (qPCR) showed better sensitivity; qPCR without pre-amplification was able to detect CTC-related genes in Parsortix PR1-enriched cells when as few as 5 SKBR3 cells were spiked into blood. Finally, the HTG EdgeSeq nuclease protection assay was able to profile mRNA expression of over 2,560 cancer-related genes from Parsortix PR1 enriched cells, showing enrichment in cancer signaling pathways and *ERBB2*, *KRT19*, and *KRT7*. Overall, the Parsortix PR1 platform may be amenable to transition into routine clinical workflows.

## Background

Liquid biopsy has the potential to serve as a surrogate for tissue biopsy. In contrast to traditional tissue biopsies, liquid biopsies are minimally invasive, relatively simple to perform, cost effective, expeditious, and easily repeatable over the course of treatment. A principal element in the paradigm of liquid biopsy is circulating tumor cells (CTCs), which are cells that have detached from primary tumors and shed into the vascular system. Every tumor evolves

**Funding:** We would like to disclose that the Angle Europe Limited provided the Parsortix® PR1 instrumentation and disposable cassettes for evaluation purposes and JMR serves on the scientific review board. Furthermore, one of the molecular assays, the QuantiGene Plex, was provided by Thermo Fisher Scientific eBioscience for evaluation purposes. No financial support was provided by HTG Molecular for the EdgeSeq gene expression assay, but MD Anderson has subsequently established a collaborative research agreement with HTG to develop a smaller version of the gene expression panel presented here.

**Competing interests:** I have read the journal's policy and the authors of this manuscript have the following competing interests: JMR serves on the scientific advisory board for Angle PLC. This does not alter our adherence to PLOS ONE policies on sharing data and materials.

differently with time and treatment, and CTCs can harbor valuable information about the characteristics of a patient's particular disease over time [1]. However, isolating the rare CTCs from an overabundance of blood elements is a complex technical challenge.

There are an increasing number of CTC enrichment platforms [2]. Because of the relative rarity of CTCs in blood, optimizing the sensitivity of these technologies is crucial. The only such technology approved by the U.S. Food and Drug Administration is the CELLSEARCH® system marketed by Menarini-Silicon Biosystems [3], which enumerates CTCs in the blood of patients with metastatic breast, colorectal, and prostate cancer [4]. CELLSEARCH is premised on an affinity-based capture of cells with surface expression of the glycoprotein epithelial cell adhesion molecule (EPCAM). EPCAM is frequently, but not ubiquitously, over-expressed by carcinomas [5]. Critically, EPCAM is down-regulated during epithelial-to-mesenchymal transition (EMT), a process necessary for cancer cells to disseminate into blood and for subsequent metastasis. Biasing the initial enrichment of CTCs by using EPCAM expression may underestimate the presence of mesenchymal CTCs with down-regulated EPCAM [6–8]. Therefore, alternative enrichment methods based on cellular properties such as size and deformability may allow characterization of a greater breadth of CTCs. The current report analyzes the sensitivity of one such technology.

The Parsortix® PR1 system (Angle Europe Limited, Guildford, United Kingdom) filters blood through a disposable microfluidics cassette containing serpentine channels with steps leading up to a plateau with a specific critical gap width designed to provide a large surface area for cell capture. Cassettes are available with critical gap widths of 4.5, 6.5, 8, or 10 µm. The cassette mechanically enriches the larger, more rigid tumor cells while allowing the smaller, more pliable cells (e.g. most red and white blood cells) to flow through [9]. After enrichment, the captured CTCs can be stained and counted *in situ* or eluted from the cassette via a flush of buffer solution and harvested for downstream analysis.

Enumeration of EPCAM positive CTCs by CELLSEARCH, while highly prognostic in multiple cancer types, has not been able to guide treatment strategies in a way that improves clinical outcomes [10]. Molecular characterization of CTCs could potentially help guide treatment strategies to improve clinical outcome. The Parsortix PR1 system, which captures CTCs without using antibodies, also bears the advantage of easy harvesting of the captured cells for subsequent downstream molecular characterization with CTC counts similar to CellSearch for cells expressing EPCAM [11, 12]. The prime objective of this study was to evaluate, optimize, and validate the isolation of breast cancer cell lines spiked into healthy donor blood (HDB) and enriched using Parsortix PR1 followed by downstream molecular characterization using different methods: QuantiGene Plex, a sensitive assay exploiting branch DNA technology (Thermo Fisher Scientific, Waltham, MA); quantitative real-time reverse transcription–polymerase chain reaction (qRT-PCR); and high-throughput genomics with the HTG EdgeSeq Assay (HTG Molecular Diagnostics, Tucson, AZ).

## Methods

### Cell culture

Five breast cancer cell lines and one lung cancer cell line were used to model CTCs (S1 Table in S1 File). The non–small cell lung cancer (NSCLC) cell line H1299 was cultured in RPMI1640 medium containing 10% fetal bovine serum and penicillin-streptomycin (100 units/mL). The hormone receptor-positive cell line MCF-7, the HER2-positive SKBR3, the triple-negative MDA-MB-453, and predominantly mesenchymal triple-negative MDA-MB-231 breast cancer cell lines were grown in DMEM/F12. The HER2-positive inflammatory breast cancer cell line SUM-190 was grown in DMEM supplemented with 5% fetal bovine serum,

5 μg/mL insulin, and 1 μg/mL hydrocortisone [13]. All cell lines were grown in an incubator at 37˚C containing 5% $CO_2$. Cells were grown to 70% confluence and then detached using 0.25% trypsin ethylenediaminetetraacetic acid (EDTA), harvested, and assessed for cell viability by trypan blue exclusion or acridine orange and propidium iodide double-staining.

## Blood draw

Written consent was obtained from healthy volunteers according to the Institutional Review Board regulations of The University of Texas MD Anderson Cancer Center as approved under IRB protocol PA14-0063 and in accordance with the Declaration of Helsinki as revised in 2008. HDB was collected by peripheral venipuncture into 10mL vacutainers coated in EDTA (BD Vacutainer 366643, Becton Dickinson, Franklin Lakes, NJ) and generally processed within 4 hours of collection.

## Cell spiking

Single cells from cultured cell lines were transferred into aliquots of HDB using a TransferMan 4r micromanipulation system (Eppendorf; Hamburg, Germany). The micromanipulator was mounted on an inverted phase contrast microscope (Nikon, Tokyo, Japan) to visualize tumor cells. For enumeration experiments, cells to be spiked were pre-labeled with fluorescent dye (CellTracker Orange CMRA dye, Thermo Fisher Scientific, Waltham, MA); unstained cells were used as controls for molecular experiments. Using a glass microcapillary tube with a 60-μm inner diameter, we individually deposited trypsinized tumor cells into a 20μL drop of culture medium on a non-adherent flat surface for enumeration. The droplet of culture medium holding the enumerated tumor cells was aspirated using a pipette and dispensed into 5mL of HDB. Following aspiration of the cells, the temporary enumeration area of the petri dish was re-examined for residual cells, and the cell count was adjusted to reflect any remaining cells not transferred. Controls for molecular studies were created using HDB from the same donor that was processed through Parsortix PR1 without the addition of cultured cells, representing an un-spiked sample. The number of spiked cells is reported per sample of blood enriched by the system, generally at least 5mL. For most experiments, cells were spiked into blood within 4 hours of phlebotomy. HDB with spiked cells was placed unto the Parsortix instrument to initiate separation generally within 1 hour of cell spiking.

## CTC enrichment by Parsortix PR1 system

The microfluidics-based Parsortix PR1 system has been described in detail by Miller et al. [14]. For recovery experiments, the GEN3D cassettes were removed from the Parsortix PR1 system after cell capture and prior to harvest. Each cassette was examined under a fluorescence microscope for review by 3 individuals in a blinded fashion to enumerate the captured cells. After counting, the cassettes were reinserted into the Parsortix PR1 system and the contents flushed into a single well of a flat-bottom 96-well microtiter plate using phosphate-buffered saline (PBS). Using fluorescence microscopy for counting, cell capture rate was defined as the percentage of cells spiked into blood that were captured in the cassette, and cell harvest rate was defined as the percentage of spiked cells that were flushed out of the Parsortix PR1 system. For linearity experiments, this process was performed in triplicate using the same HDB donor or the same cell line to minimize variability due to different donors. The Parsortix PR1 maintains constant pressure, therefore separation time is dependent upon blood volume and latent sample-specific factors such as blood viscosity. The instrument processes a typical 10mL tube of HD EDTA peripheral blood trough enrichment and harvest in 2.3 hours (mean 2 hours 18 minutes, standard deviation of 37 minutes).

## Gene expression by QuantiGene Plex assay

Following separation, the Parsortix PR1 system elutes enriched cells in 210μL of PBS. For the QuantiGene assay, this entire volume of harvested cells was subjected to cell lysis by the addition of 90μL of lysis buffer, for a final volume of 300μL, without the use of centrifugation to concentrate the cells. From this lysate, 80μL was added to each of the duplicate wells in a QuantiGene Plex Assay (Thermo Fisher Scientific, Waltham, MA). This assay was used for multiplexing to simultaneously detect the transcripts of 5 epithelial genes (*CDH1*, *EGFR*, *ERBB2*, *KRT18*, *MUC1*) and 20 CTC-related and/or breast cancer-related genes, including mesenchymal genes (*CDH2*, *FN1*), cancer stem cell-related genes (*ALDH1A1*, *CD44*), genes from key cancer signaling pathways (*AR*, *CTNNB1*, *ESR1*, *ESR2*, *FAS*, *FASLG*, *FOXO3*, *GATA3*, *IGF1R*, *STAT3*), growth factor genes (*PDGFRB*, *TGFB1*, *VEGFA*), and reference genes (*GUSB*, *HPRT1*, *TBP*). See S2 Table in S1 File for gene abbreviations. The assay was performed in a 96-well plate, and the signal was detected using a Luminex LX100 Microplate reader (Luminex, Austin, TX) according to the manufacturer's instructions. A gene was considered detectable if the transcript level was 2.5 standard deviations above the mean transcript level of the gene in 4 sham-spiked HDB samples. Human Universal RNA was included as a technical control. To assess linearity and precision, cells were spiked into HDB at levels spanning a 3-log range, including 50, 500 and 5000 spiked cells per 5mL of blood. To measure sensitivity, we quantified gene expression in cells harvested from spiked and sham-spiked HDB samples using the Parsortix PR1. Expression levels were considered positive if the non-normalized expression (as measured by mean fluorescence intensity) for a given gene was 2.5 standard deviations above the mean for sham-spiked samples for the same gene, measured from 4 donors.

## Gene expression by qRT-PCR with pre-amplification using SYBR-PrimePCR assays

In pursuit of higher sensitivity, we also performed qRT-PCR on lysates of the harvested cells. Cells were independently spiked into HDB at 100 cells per 5mL of HDB and 50 cells per 5mL of HDB, with sham-spiked samples as controls. The breast cancer lines MCF-7, MDA-MB-453, SUM-190 and MDA-MB-231were spiked into separate HDB samples and enriched using Parsortix PR1. The harvest lysate was subsequently subjected to qRT-PCR quantification. Fifteen target genes and 3 housekeeping genes were included in the PrimePCR SYBR Green Assay panel from Bio-Rad Laboratories (Hercules, CA). The panel of 15 target genes consisted of epithelial or breast-specific genes (*CDH1* [E-cadherin], *EGFR* [ERBB1], *ERBB2* [HER2], *EPCAM*, *MUC1*, *KRT5*, *KRT8*, *KRT18*, *SCGB2A2* [mammaglobin-A]), mesenchymal markers not highly expressed in blood (*CDH2* [N-cadherin], *VIM*, *ZEB2*), and several markers for characterization of enriched CTCs (*ESR1*, CD274 (*PDL1*), *SRC*). Preamplification was performed using gene-specific primers for 10 cycles according to the manufacturer's instructions. SYBR green–based qRT-PCR amplification was performed in an ABI 7900HT Fast Real-Time PCR system (Applied Biosystems, Thermo Fisher Scientific) according to the manufacturer's instructions (Bio-Rad Laboratories). Quantification cycle (Cq) values of ≥35 were rejected. Three housekeeping reference genes (GAPDH, HPRT, B2M) were used to normalize gene expression [15, 16], and *PTPRC* (CD45) was included as a white blood cell control gene.

## Gene expression by qRT-PCR without pre-amplification using probe assays

Independent cell line spiking experiments using SKBR3, each with a different normal donor, with independent transcript quantifications were performed over a 2-week period. The

harvested cells in 210µL of PBS was centrifuged and reduced to about 20µL before lysis with 330µL Qiagen RLT buffer. RNA was enriched using RNeasy Plus kits (Qiagen, Hilden, Germany) and transcribed using ABI High-Capacity cDNA Reverse Transcription Kits (Thermo Fisher Scientific). A small panel of 9 genes was amplified using PrimePCR probes and Bio-Rad SsoAdvanced Universal Probes Supermix (Bio-Rad Laboratories) on an ABI 7900HT Fast Real-Time PCR system according to the manufacturer's instructions. Assays measured the following genes: *GAPDH* (Unique Assay ID: qHsaCEP0041396) and *B2M* (qHsaCIP0029872) as housekeeping genes; red blood cell marker *GYPA* (qHsaCEP0057766); epithelial genes *EPCAM* (qHsaCEP0051089), *KRT19* (qHsaCEP0054049), and *ERBB2* (qHsaCEP0052301); mesenchymal genes *TWIST* (qHsaCEP0051221) and *SNAI2* (qHsaCIP0027976); and white blood cell marker *PTPRC* (CD45; qHsaCEP0041630).

## Gene expression profiling using HTG EdgeSeq assay

The HTG EdgeSeq Assay (HTG Molecular Diagnostics) is an extraction-free, next-generation sequencing-based RNA expression assay for difficult sample types. SKBR3 breast cancer-derived cells (0, 10, 50, and 500 cells) were spiked into 5mL aliquots of HDB and processed through the Parsortix PR1 system. The harvested cells were subjected to direct lysis using 2X lysis buffer provided by the manufacturer. A neat, positive control of 500 SKBR3 cells not spiked into HDB was included in the HTG assay. An EdgeSeq PATH panel with 470 cancer genes was used for this experiment according to the manufacturer's instructions (HTG Molecular Diagnostics). After EdgeSeq capture and PCR amplification of mRNA, samples were pooled and analyzed with an Illumina MiSeq system and parsed with HTG software. Data were analyzed through the use of IPA (QIAGEN Inc., https://www.qiagenbioinformatics.com/products/ingenuity-pathway-analysis).

## Results

To simulate CTCs in patient peripheral blood, live cells from established cancer-derived cell lines were spiked into HDB. These contrived samples were then subjected to CTC enrichment using the Parsortix PR1 system.

## CTC enrichment by Parsortix PR1

**Optimization of critical gap size.**   To determine the optimal cassette configuration for the highest capture level of CTCs by the Parsortix PR1 system, we evaluated cassettes with 4 different critical gap sizes (4.5, 6.5, 8, and 10 µm) using 2 breast cancer-derived cell lines and 1 NSCLC-derived cell line. SKBR3, a frequently used CTC model, was selected to represent HER2-positive breast cancer; MDA-MB-231 cells were selected to represent triple-negative breast cancer with partial mesenchymal features; and H1299 cells were selected to represent non-small cell lung cancer. A total of 12 runs of 5mL HDB samples encompassing 3 normal donors was performed, each sample spiked with 100 live tumor cells. Capture rates by the Parsortix PR1 system increased as the cassette gap size decreased (Table 1). The lowest capture rate by the Parsortix PR1 system was observed with the 10µm gap cassette followed by the 8µm gap cassette. The capture rates of the Parsortix PR1 system were highest with the 4.5-µm gap size cassettes but these capture rates were accompanied by a relatively high levels of other blood cells, making enumeration difficult. Hence, the 6.5-µm gap size cassette was chosen as being optimal for cell capture and ease of enumeration.

**Assessment of cell capture and harvest rates.**   Using the optimal cassette gap size, we established cell capture, harvest, and recovery rates for the Parsortix PR1 system using SKBR3, MDA-MB-231, and H1299 cells spiked into HDB. Three individuals counted the numbers of

**Table 1. Capture percentage across cell separation cassette sizes.**

| Cassette Size | Cell Lines | | |
|---|---|---|---|
| | **SKBR3** | **MDA-MB-231** | **H1299** |
| | Epithelial | Mesenchymal | Epithelial |
| 4.5 µm | 69.90 | 55.45 | 90.59 |
| **6.5 µm** | 69.61 | 66.67 | 51.00 |
| 8.0 µm | 50.25 | 35.58 | 58.42 |
| 10 µm | 42.96 | 13.07 | 41.50 |

A critical cassette gap size of 6.5 µm was optimal for enrichment.

cells captured and harvested in a blinded fashion, and the data were summarized as a stacked, clustered column graph (Fig 1A).

Of note, the Parsortix PR1 system is capable of capturing and harvesting a single cell spiked into a 5mL sample of blood. In experiments with SKBR3 cells spiked in HDB, we were able to demonstrate the capture of a single cell by the Parsortix PR1 system in 2 of the 3 samples. MDA-MB-231 cells in HBD consistently had the lowest rate of capture, possibly because these cells possess some mesenchymal features that may make them more prone to migrate and more likely to escape from the microfluidics cassette.

**Assessment of linearity.** Each of 4 HDB samples (5mL each) was spiked with a predetermined number of cells (0, 10, 50, or 100 cells) from each of 3 cancer-derived cell lines (H1299, SKBR3, MDA-MB-231) using the micromanipulator and then enriched for CTCs using the

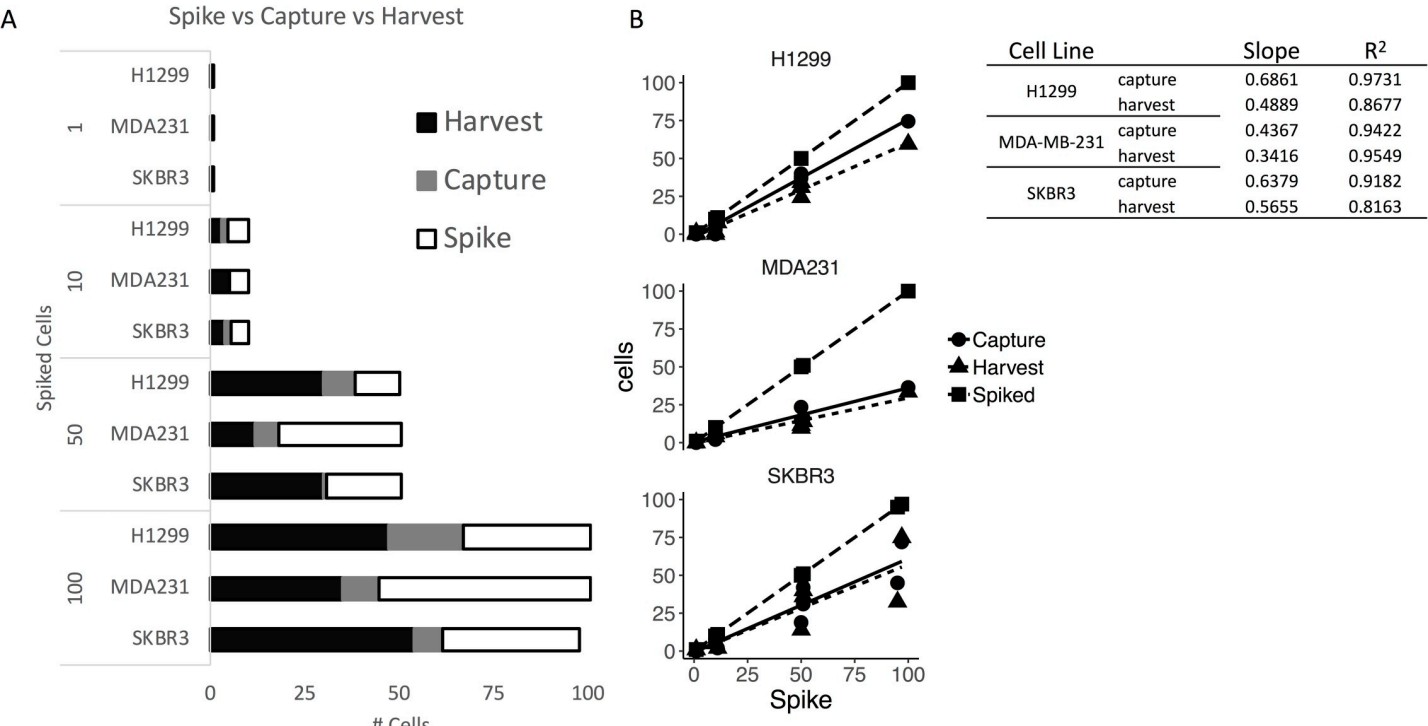

**Fig 1. Capture and harvest efficiency and linearity of Parsortix PR1 by imaging.** A): The mean total number of cells spiked (white), captured (gray), and harvested (black) from healthy donor blood (n = 3 normal donors). B): Scatter plots with linear regressions performed on the triplicate data. Each graph shows a linear regression for the total number of cells (square), the number captured (circle), and the number harvested (triangle). These plots show the linearity of the data.

Parsortix PR1 system. The linearity of cell capture and harvest was characterized for all 3 cell lines (Fig 1B), with high $R^2$ values observed for both capture and harvest. These results indicate a strong linear relationship between the number of target cells spiked and recovered, and suggests that a count of captured CTCs in a patient sample would reflect the relative frequency of CTCs in the patient's blood.

The slope coefficients for MDA-MD-231, H1299, and SKBR3 cell capture were 0.4367, 0.6861 and 0.6379, respectively. The slope coefficients for the cell harvests were 8–20% lower, indicating a small loss of cells between capture and harvest. Overall, approximately half of all target cells in a sample are captured and harvested, with a small loss during harvest. This pattern can be seen directly in the best-fit lines (Fig 1B), as the slopes of the capture and harvest lines are roughly half the magnitude of the slopes of the spike lines.

**Molecular characterization of enriched CTCs.** Compared with imaging, multiplex gene expression analysis quantifies a greater breadth of cellular characteristics and is more amenable to higher-throughput screening methods. For example, mesenchymal markers such as *TWIST*, *VIM*, and *ZEB2*, which are low in blood elements but high in some *EPCAM*-negative CTCs, can be quantified to identify CTCs that would not be enriched using *EPCAM*-based capture strategies. We tested the ability of different gene expression profiling modalities to detect and molecularly characterize model CTCs enriched from HDB using the Parsortix PR1 system, starting with the QuantiGene assay.

QuantiGene RNA assays are hybridization-based assays that use a branched DNA technology for signal amplification for the direct quantitation of gene expression transcripts. These assays are amenable to using any cell material directly without RNA purification, thereby avoiding variability introduced by human error and loss of material due to sample processing. This ability is especially significant for liquid biopsy products owing to the dearth of available cells for downstream molecular analysis. For this study, 25 genes were analyzed, including transcripts representative of epithelial cells, mesenchymal cells, cancer stem cells and other tumor-related pathways.

**Specificity, linearity and precision of QuantiGene RNA assay.** To test the specificity of gene expression analysis for CTCs, 5,000 target cells were spiked into HDB, which was then processed using the Parsortix PR1 system to capture and harvest the spiked cells. The harvested cells were interrogated for epithelial gene expression from each of 4 cell lines harvested from HDB using 1/4 of the cell lysate, which would represent approximately 1,250 spiked cells. In HDB samples spiked with SUM190 and MCF-7 cells, 4 of 5 epithelial genes measured were detectable; in MDA-MB-453–spiked HDB samples, 2 of 4 epithelial genes known to be expressed in them were detectable; and in MDA-MB-231–spiked HDB samples, 3 of 3 epithelial genes known to be expressed in them were detectable (Table 2). Furthermore, the expected

**Table 2. QuantiGene branched DNA assay.**

|  |  | SUM190 | MCF-7 | MDA-MB-453 | MDA-MB-231 |
|---|---|---|---|---|---|
| Transcripts | CDH1 | + | + | − | N/A |
|  | KRT18 | + | + | + | + |
|  | ERBB2 | + | + | + | + |
|  | EGFR | − | + | − | + |
|  | MUC1 | + | − | N/A | N/A |

For epithelial genes known to be expressed by these cells, good correlation was seen between expected and observed gene expression using branched DNA following spiking of 5000 tumor cells into 5 mL of blood and separation by Parsortix PR1 system: 4 of 5 genes were detected in SUM190 and MCF-7 cells, and 2 of 4 genes were detected in MDA-MB-453 cells, 3 of 3 genes were detected by MDA-MB-231. +, detected; N/A, not typical for this cell line; −, expected but not detected.

genes that were not detected in enriched cell–spiked samples (EGFR in SUM190, MUC1 in MCF-7, EGFR and CDH1 in MDA-MB-453), were all detectable at very low levels with these primers in neat, unenriched cells of the respective cell lines).

To assess linearity and precision of molecular techniques for CTCs enriched by the Parsortix PR1 system, live cells were spiked into HDB at levels spanning a 2-log range including 50, 500 and 5000 spiked cells per 5mL of blood. Gene expression was measured with the QuantiGene assay, and expression levels were positively correlated with the number of cells spiked into HDB, with the correlation maintaining good linearity over the range. Most genes tested had an $R^2$ above 0.90 (Fig 2A), and coefficients of variance (%CVs) for most replicates were under 10%.

To measure sensitivity, we quantified gene expression in spiked and sham-spiked HDB samples enriched by the Parsortix PR1 system. Expression of the epithelial gene KRT18 was positive from as few as 50 spiked cells from the breast cancer cell lines SUM190 (basal) and MCF-7 (luminal). The SUM190 cell line had several genes expressed when >50 cells were spiked into HDB. Gene transcripts from MDA-MB-453 spiked cells were detected only with cell spikes of 500 cells or more for the 25 genes tested, including genes associated with epithelial phenotypes, mesenchymal phenotypes, and cancer stem cell phenotypes. Gene transcripts from the partially mesenchymal cell line MDA-MB-231 were only detected when 5000 cells were spiked into HDB (Fig 2B).

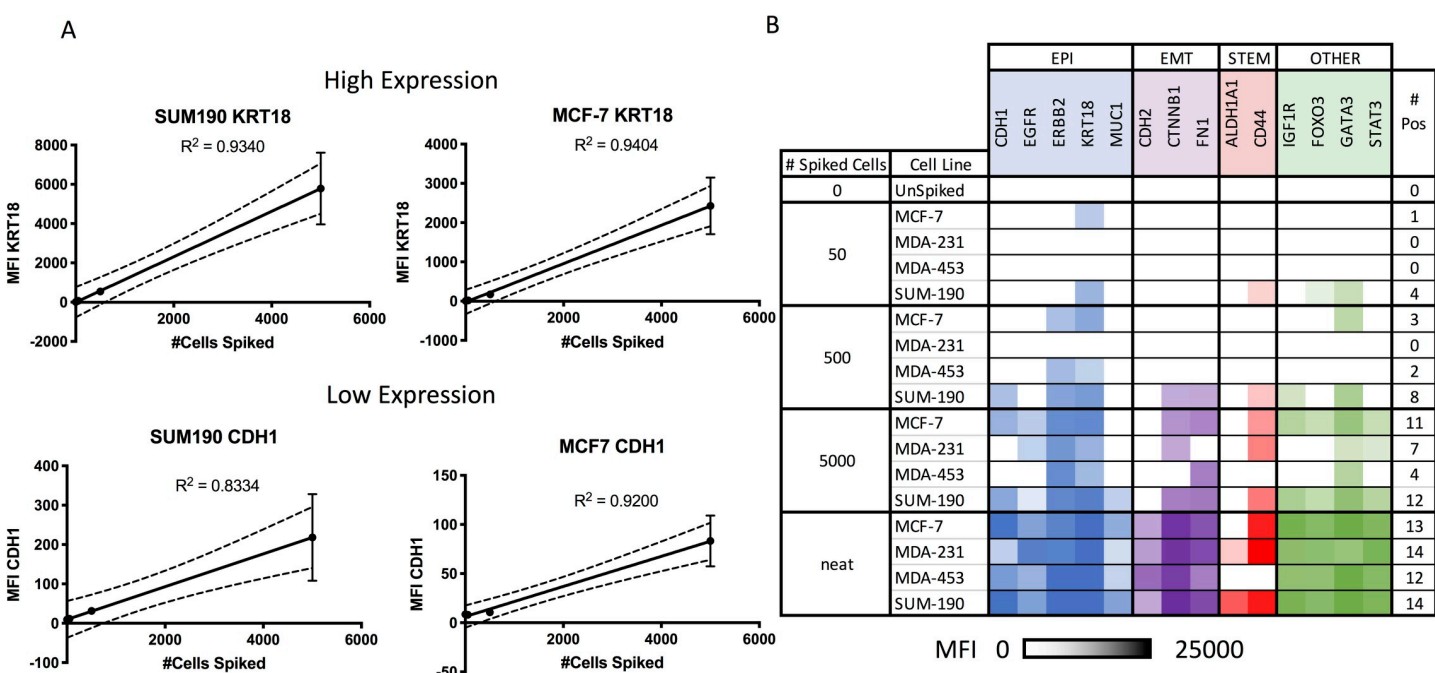

**Fig 2. Linearity and sensitivity of QuantiGene branched DNA gene expression without amplification after Parsortix PR1 enrichment.** Cells of the specified cell line were spiked into healthy donor blood, enriched by Parsortix PR1 system, and lysed. One quarter of the lysate from each sample was hybridized in duplicate. Gene expression of neat cultured cells, not spiked into blood, were used as controls. A) Linearity: Expression levels correlated well with the number of cells spiked into healthy donor blood, and this correlation was maintained ($R^2 > 0.9$) for most of the 25 genes tested. Representative data from tests of 2 genes (low and high expression) in 2 cell lines are shown. B) Sensitivity: *KRT18* gene transcripts were detected in healthy donor blood spiked with as few as 50 SUM-190 cells or MCF-7 cells. Several gene transcripts were detected when >50 cells were spiked. MDA-MB-453 gene transcripts were detected only in cell spikes of 500 cells or higher. Gene transcripts were detected in the partially mesenchymal cell line MDA-MB-231 only when several thousand cells were spiked. Numbers of cells spiked are specified here (the numbers of cells captured and therefore subjected to expression analysis will be lower). The gene expression from neat cells represents about 8000 cultured cells per assay well. Heat map shows natural log transformed expression after subtraction of the sham-spiked HDB (mean fluorescence intensity [MFI]) above the mean+ 1 standard deviation of un-spiked, enriched samples.

**Gene expression by qRT-PCR with and without pre-amplification.** Because of the relatively low sensitivity of gene expression profiling without amplification of the target nucleic acid sequences, we also evaluated the ability to detect enriched cell spikes using qRT-PCR preceded by pre-amplification. For the breast cancer lines MCF-7 (hormone receptor positive), MDA-MB-453 (HER2 positive), SUM-190 (inflammatory breast cancer, HER2-positive) and MDA-MB-231 (partial mesenchymal, triple-negative breast cancer), as the number of spiked cells into HDB increased, the number of gene transcripts detected also increased (S1 Fig in S1 File). In contrast to the QuantiGene branched-DNA method, qRT-PCR was able to detect as few as 50 of the partially mesenchymal MDA-MB-231 cells spiked into HDB with a threshold of 2 times greater expression compared to sham-spiked HDB samples (Fig 3A), suggesting better sensitivity.

Since we observed better sensitivity when the target sequence was pre-amplified before use of the PrimePCR system, we developed a smaller primer set that could be used without pre-amplification and the associated biases. In reproducibility experiments, with a panel of 9 genes, we were able to reliably detect CTC-related genes in enriched samples from multiple donors containing as few as 5 spiked SKBR3 cells (Fig 3B and 3C). Although quantified at very low expression levels with a Cq of 35 or higher, *EPCAM* expression from 5 spiked SKBR3 cells was consistently higher than that of matched sham-spiked blood. Expression of the epithelial genes *ERBB2* and *KRT19* in enriched blood containing 5 spiked SKBR3 cells was clearly positive, although low levels of *ERBB2* expression were detected in the sham-spiked HDB samples. The housekeeping genes *B2M* and *GAPDH* were fairly constant since background enrichment

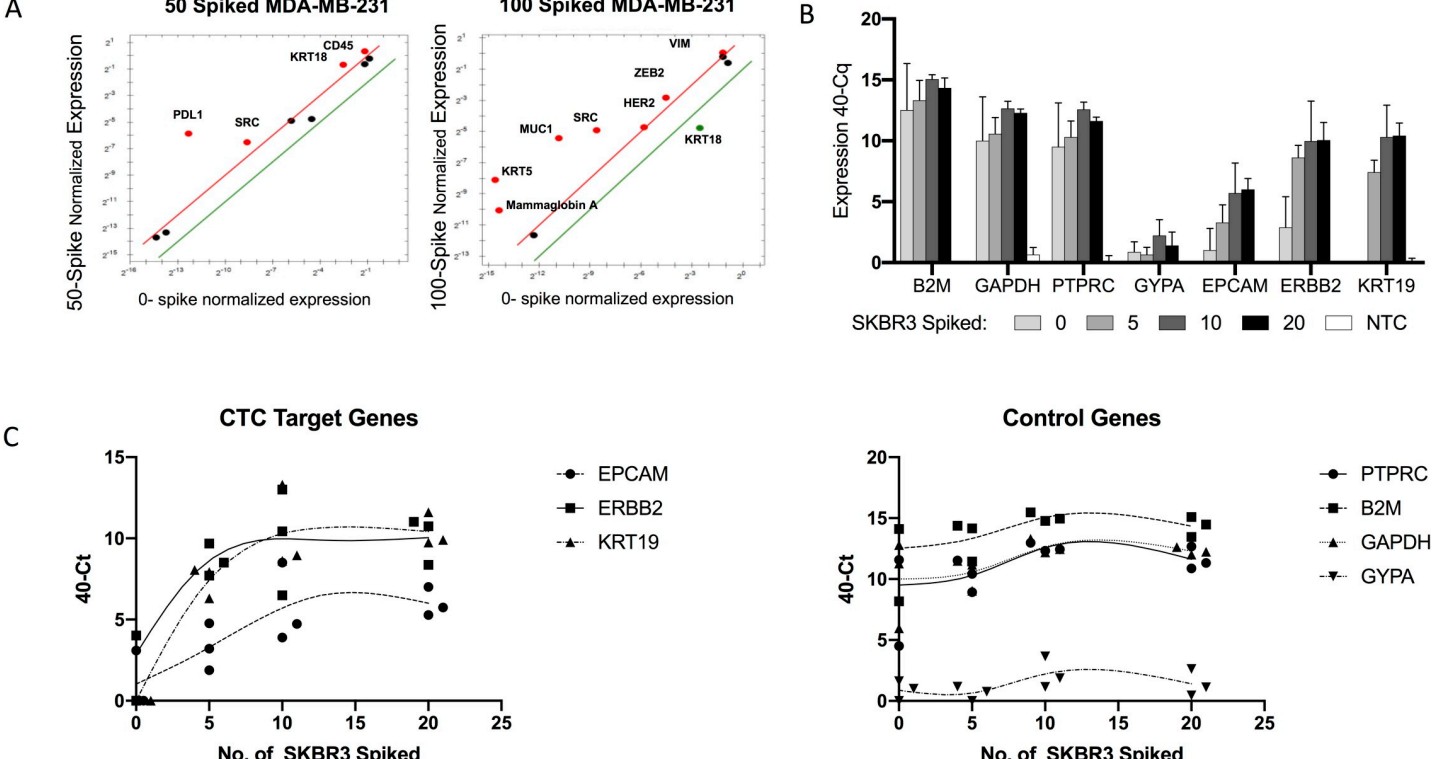

**Fig 3. Gene expression by qRT-PCR after Parsortix PR1 enrichment.** A) Using pre-amplification, qRT-PCR can detect MDA-MB-231–related genes in an enriched sample of 50 cells spiked into ~5 ml HDB. B,C) A panel of 9 genes was tested on independent spiked samples over a 2-week period using a smaller gene panel with no pre-amplification. Healthy donor blood was spiked with 0, 5, 10, or 20 SKBR3 cells prior to enrichment with Parsortix PR1 system.

of white blood cells was the dominant factor. However, quantitating a 1-Cq difference (i.e., a doubling of the number of cells) approaches the limits of the system.

**Molecular profiling with HTG EdgeSeq.** Because of the sensitivity issues described, as a proof of concept, we evaluated the HTG EdgeSeq system for molecular profiling of CTCs using the SKBR3 breast cancer cell line. To test the ability of HTG EdgeSeq to detect gene expression of cell lines, we evaluated HDB samples spiked with 0, 10, 50, or 500 SKBR3 cells after enrichment using the Parsortix PR1 system. Expression levels of *ERBB2*, *KRT19*, and *KRT7* were significantly higher in a pairwise comparison of sham-spiked HDB samples and HDB samples spiked with 50 and 500 SKBR3 cells combined (P < 0.05 by student t-test, log2-normalized counts). In addition, two genes that were not over-expressed by neat SKBR3 cells, *MPO* and *ALB*, showed higher expression in Parsortix PR1 harvests of spiked SKBR3 cells. However, these two genes exhibited low expression with a small absolute change upon enrichment that was statistically significant. This result could be related to the interaction of SKBR3 cells with mismatched hematopoietic cells, but was not examined further since it was beyond the scope of this study (S2A Fig in S1 File). Genes with the highest fold-changes in expression in the spiked HDB samples compared to sham-spiked HDB samples were the genes also over-expressed by neat SKBR3 cells (S2B Fig in S1 File). The 30 highest-expressed genes out of the 470 genes in the HTG PATH panel are listed for each sample in S3 Fig in S1 File. *KRT18* was easily detected in the harvests from HDB samples spiked with 10 SKBR3 cells. As the number of spiked SKBR3 cells in HDB increased, more epithelial genes and proliferation-related genes were detected. The 10 highest-expressed genes, including the CTC markers *S100A6*, *ERBB2*, *KRT18*, *KRT8*, *KRT19*, and *KRT7*, were the same for the 500 neat SKBR3 cells and the HDB samples spiked with 500 SKBR3 cells and processed through the Parsortix PR1 system. The highest-expressed genes in the SKBR3-spiked HDB samples compared with the sham-spiked HDB enriched samples included *ERBB2*, *KRT7*, *HSPB1*, and *KRT19*; each of these transcripts was at least 60 times higher in the HDB sample spiked with 100 SKBR3 cells, 20 times higher in the sample spiked with 50 SKBR3 cells, and 2 times in the sample spiked with 10 cells.

To identify a gene signature for SKBR3 cells that could be detected in spiked HDB samples, we characterized neat SKBR3 cells using HTG PATH assay for gene expression. Of the 470 genes in the PATH panel, 87 genes were expressed at a level of more than 2.5 standard deviations higher in the neat SKBR3 cell sample compared to the sham-spiked HDB sample when normalized to counts per million reads. These 87 genes accounted for over half of the PATH panel expression in the SKBR3 cells, with a cumulative total of 529,733 reads per million. From this signature, 85 genes (98%) were positive in the Parsortix PR1 harvests from HDB samples spiked with 500 SKBR3 cells (Fig 4A). *KRT18* and *KRT8* were the only 2 genes with a %CV of >25% in the sham-spiked enriched samples; therefore, they were also included in the signature even though their enriched expression was less than 2.5 standard deviations higher in the spiked HDB samples compared to the sham-spiked HDB samples.

In Fig 4B, a clear negative correlation appears between the cluster of genes that are molecular markers of CTCs and the cluster of genes that are more specific to blood elements. Furthermore, Ingenuity Pathway Analysis (S4 Fig in S1 File) showed that the top 5 pathways enriched in the HDB samples spiked with 50 and 500 SKBR3 cells compared with the sham-spiked HDB samples were ovarian cancer signaling, hereditary breast cancer signaling, mismatch repair in eukaryotes, endometrial cancer signaling, and glioma signaling (all $P < 0.5 \times 10^{-9}$), and the top upstream regulators included *ERBB2*, *CDKN1A*, estrogen receptor, and *TP53* (S4 Fig in S1 File). Additionally, the top disease pathways expressed at higher levels in the spiked samples compared with sham-spiked HDB samples were cell viability, cell survival, proliferation of epithelial cells, growth of epithelial cells, and cell movement (which includes molecules that inhibit movement, such as KRT19). Together, these pathway data suggest that cancer cells

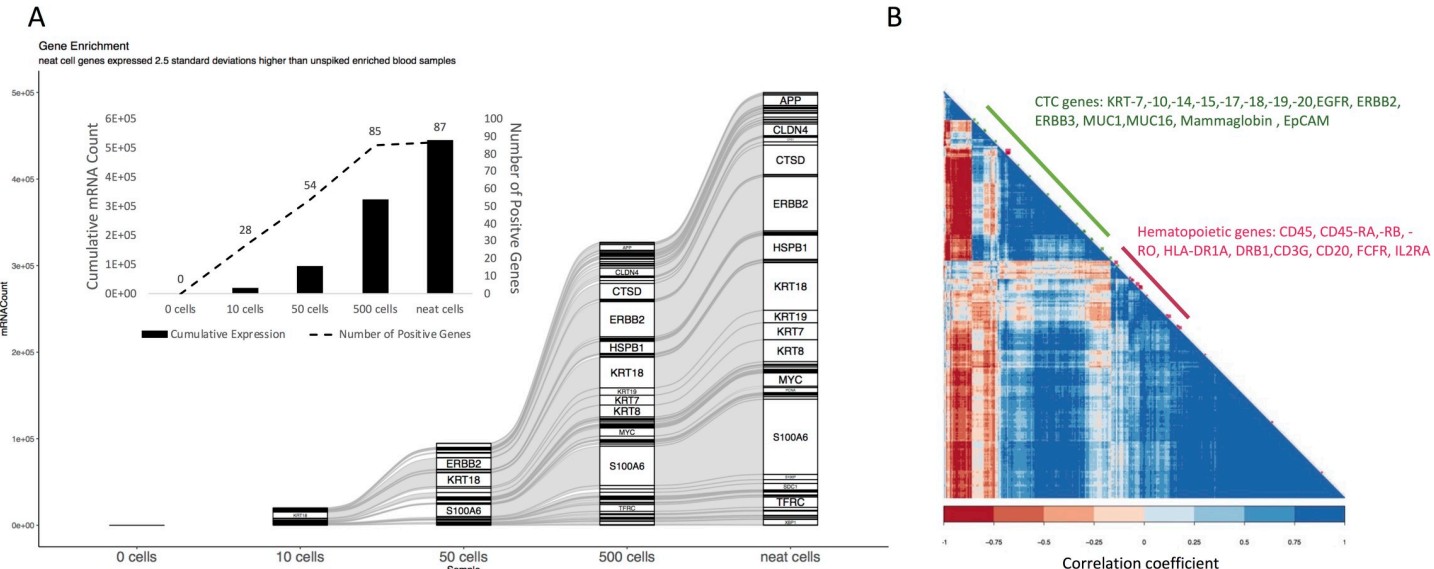

**Fig 4. Molecular profiling by HTG after Parsortix PR1 enrichment.** Healthy donor blood spiked with 0, 10, 50, or 500 SKBR3 cells was subjected to gene expression profiling with the HTG EdgeSeq PATH assay. A sample of 500 neat SKBR3 cells, with no blood components, was used as a control. A) Expression levels as mRNA counts per million reads are shown for each gene expressed at levels 2.5 standard deviations higher than in the sham-spiked enriched samples. Epithelial genes including *ERBB2* and *KRT18* are expressed by neat SKBR3 cells and are decreasingly common as the number of cells spiked into blood decreases prior to Parsortix PR1 enrichment and HTG assessment. The insert shows the cumulative transcript counts and total number of genes from the SKBR3 signature above the cut-off for each sample. B) Heatmap of Pearson correlation analysis shows that epithelial genes and hematopoietic genes are generally clustered as disparate groups and are inversely related.

can be detected by gene expression analysis using the HTG EdgeSeq system following enrichment from blood using the Parsortix PR1 system.

## Discussion

Here, we validated the ability of the Parsortix PR1 CTC enrichment platform to enrich small numbers of cultured cells spiked into HDB using several downstream analyses including molecular characterization. Using pre-labeled, live cultured cells, we determined that the Parsortix PR1 system is capable of enriching lung and breast cancer-derived cell lines as well as EMT/mesenchymal-like cells, and that the majority of the cells captured by the separation cassette can be harvested. The Parsortix PR1 system was shown to be capable of capturing and harvesting a single cell spiked into 5mL of blood, which is encouraging given the recent data suggesting that detection of even a single CTC can be prognostic during adjuvant treatment of breast cancer [17].

Previous validation studies of the Parsortix PR1 system have shown similar recovery of epithelial cells [11, 18–21]. The imperfect capture efficiency can be partially attributed to deformation and capillary flow required by the platform resulting in lysis of some CTC. Although we have observed ruptured cells on Cytospin preparations of enriched CTC, quantifying this effect was beyond the scope of this manuscript. The reciprocal relationship between capture efficiency and cassette gap size is expected for a size-based system. Any population of cells has a distribution of cell sizes; cells smaller than the gap size will pass through the system without enrichment [11]. A larger portion of this distribution is lost to capture as the gap size threshold is increased. In addition and similar to others [12], we show that the Parsortix PR1 system can enrich partially mesenchymal cells with low EpCAM expression, such as MDA-MB-231 cells, although they were captured less efficiently compared to cells of epithelial origin. This result contrasts standard EpCAM-based capture systems such as the CELLSEARCH CTC Test.

Using such systems, we had previously seen no enrichment of cells with low EPCAM expression [6]. The role of EPCAM in CTC dissemination, aggressiveness and EMT is poorly understood, as EPCAM both promotes homotypic adhesions and inhibits E-cadherin–mediated interactions [22] to affect cell signaling, migration, proliferation, and stemness [23, 24]. We and others have previously reported that mesenchymal CTCs have prognostic significance [25–27] and particular metastatic propensities [28]. In contrast, in at least one study of prostate and breast cancer patients, only the EPCAM-high and not the EPCAM-low cells were related to survival [29], while others have found that the prognostic value of mesenchymal CTCs is subtype specific [30]. The epithelial-mesenchymal spectrum of CTCs seems highly plastic and can change spatially in the bloodstream [31] and temporally with treatment, with a large portion of CTCs displaying a hybrid epithelial and mesenchymal state [32]. Single cell analysis of Parsortix-enriched CTCs have shown populations of CTC can co-express epithelial and mesenchymal markers [33]. Therefore, an antigen-agnostic enrichment approach, such as the one demonstrated here with the Parsortix PR1, may broaden the range of clinically relevant cells available for further analysis. Future clinical validation is warranted.

While enumerating CTCs from blood can provide prognostic value, the information from this type of assessment alone is limited. Phenotypic analysis alone cannot always distinguish CTCs from normal circulating epithelial cells. In addition, the reliance of enumeration on epithelial markers limits the ability to enumerate EMT CTCs. Therefore, we further characterized enriched samples by gene expression analysis using 3 modalities: 1) signal amplification by the QuantiGene Plex Assay, 2) target amplification by qRT-PCR with and without pre-amplification, and 3) high-throughput target amplification by the HTG EdgeSeq Assay. QuantiGene, which uses branch DNA to amplify multiplex target sequences hybridized to Luminex beads, was amenable to profiling large numbers of spiked cells, but it lacked sufficient sensitivity to quantify low numbers of CTCs. At 50 spiked cells, expression of target genes was only slightly higher than in HDB (Fig 2). In contrast, amplification of target sequences by qRT-PCR showed greater sensitivity. With a pre-amplification method, we were able to detect several genes from only 50 spiked MDA-MB-231 cells, despite the having a low recovery (Fig 1). However, non-specific signals with pre-amplification are well documented, including from Parsortix PR1-enriched samples [34]. Further, we could consistently detect genes from as few as 5 spiked SKBR3 cells (Fig 3).

In addition to PCR, we used the HTG EdgeSeq platform to demonstrate the ability to molecularly profile enriched CTC. This assay is especially advantageous when the quantity and quality of available biological material is limited. Similar to the QuantiGene assay, the chemistry is extraction-free and thereby reduces data bias due to RNA extraction and the accompanying sample loss. The HTG PATH Assay is designed for retrospective gene expression profiling to assess mRNA expression of 470 targets in formalin-fixed paraffin-embedded tissue samples. Compared with a larger panel such as HTG's oncology biomarker panel with 2560 genes, the smaller 470-gene PATH panel should have higher sensitivity since each of the target genes represents a larger portion of the total sequence reads, making a smaller panel a better option when sample input is limited. The panel includes proteins routinely tested in clinical pathology laboratories while providing a workflow and data output amenable to a clinical setting. Indeed, the HTG EdgeSeq DLBCL Cell of Origin Assay and next-generation sequencing [35] has obtained European Community in vitro diagnostic (CE-IVD) marking for diagnostic use.

Since CTCs are rare, repurposing harvested cells so that multiple data modalities can be obtained from a single sample would be highly advantageous. To this end, HTG has validated their gene expression panel on tumor formalin-fixed paraffin-embedded slides previously stained for hematoxylin and eosin assessment [36]. While not tested here, this capability for repurposing samples would theoretically allow bulk gene expression profiling of CTCs after

they have been counted and could mitigate the variance in CTC numbers between concurrently drawn tubes of blood. Such dual-purposing of cells is a common theme in CTC research [37, 38], but the methods are often tedious and highly research oriented. Staining followed by the HTG EdgeSeq Assay could offer a compromise between single-cell sequencing of CTCs with its high cost, uncertainty, and tedious workflow and gene expression profiling of bulk CTCs where CTC enumeration needs to be extrapolated from a separate blood tube.

The debate surrounding the fluid definition of CTCs has only intensified. Enumeration is highly operator dependent, as was noted in even the earliest reports with CELLSEARCH reports [4]. For example, using single-cell copy number alterations, Chemi et al. have shown that cells with a CTC phenotype (EpCAM-positive, CK-positive, CD45-negative) can be genomically normal [39], while Reduzzi and colleagues have shown that dual-positive cells (CK-positive, CD45-positive) may represent hybrid fusions of tumor and stromal cells[38]. Indeed, previous studies have shown that hybrids resulting from the fusion of tumor cells and stromal cells can promote metastasis [40, 41]. Such analyses show that imaging alone may not be sufficient to characterize CTCs and that multimodality characterization can be highly advantageous in clinical decision-making. Here, we demonstrate the possibility of multimodal characterization of enriched CTC. Further development and validation are needed, especially in terms of repurposing limited samples.

## Conclusions

Overall, the Parsortix PR1 system can capture a variety of spiked tumor cell lines from blood, independent of surface antigens, with captured cells amenable to downstream molecular characterization. Using 3 gene expression analysis methods, we detected several gene transcripts at high levels of sensitivity, from as few as 5 spiked cells or less. In addition, we observed a linear correlation between the number of cells spiked and quantities of RNA transcripts. As molecular characterization is gaining importance in the development of minimally-invasive clinical diagnostics, our observations have potential use in liquid biopsy.

## Supporting information

**S1 File.**
(PPTX)

## Acknowledgments

The authors would like to thank Sanda Tin her technical help, Sarah Bronson from Scientific Publications, Research Medical Library at The University of Texas MD Anderson Cancer Center (Houston, TX) for editing the manuscript, and Stefan Jellbauer from Thermo Fisher and Yong Lee from HTG Molecular Diagnostics, Inc. for their help designing assays.

Thermo Fisher Scientific eBioscience provided the QuantiGene Plex Assay for evaluation purposes. Angle PLC provided the Parsortix® PR1 instrumentation and GEN3D cell separation cassettes for evaluation purposes.

Presented in part at the American Association for Clinical Chemistry Annual Conference 2016, Philadelphia, PA.

## Author Contributions

**Conceptualization:** James M. Reuben.

**Data curation:** Evan N. Cohen, Gitanjali Jayachandran, Max R. Hardy, Ananya M. Venkata Subramanian, Xiangtian Meng.

**Formal analysis:** Evan N. Cohen, Gitanjali Jayachandran, Max R. Hardy.

**Funding acquisition:** James M. Reuben.

**Investigation:** Evan N. Cohen, Gitanjali Jayachandran, Max R. Hardy, Ananya M. Venkata Subramanian, Xiangtian Meng, James M. Reuben.

**Methodology:** Evan N. Cohen, Gitanjali Jayachandran, James M. Reuben.

**Project administration:** James M. Reuben.

**Resources:** James M. Reuben.

**Supervision:** James M. Reuben.

**Validation:** Gitanjali Jayachandran.

**Visualization:** Evan N. Cohen, Gitanjali Jayachandran, Max R. Hardy.

**Writing – original draft:** Evan N. Cohen, Gitanjali Jayachandran, James M. Reuben.

**Writing – review & editing:** Evan N. Cohen, Gitanjali Jayachandran, Max R. Hardy, James M. Reuben.

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
