## [Decision Letter · Decision Letter 0]

13 Aug 2020

PONE-D-20-19549

Antigen-agnostic microfluidics-based circulating tumor cell enrichment and downstream molecular characterization

PLOS ONE

Dear Dr. Cohen,

Thank you for submitting your manuscript to PLOS ONE. After careful consideration, we feel that it has merit but does not fully meet PLOS ONE’s publication criteria as it currently stands. Therefore, we invite you to submit a revised version of the manuscript that addresses the points raised during the review process.

Normally I require at least two reviews; however, I respect this review and they have very good points.  Please attempt to address them.

We look forward to receiving your revised manuscript.

Kind regards,

Jeffrey Chalmers, Ph.D.

Academic Editor

PLOS ONE

Journal Requirements:

"Written consent was obtained from healthy volunteers according to the Institutional Review Board regulations of The University of Texas MD Anderson Cancer Center and in accordance with the Declaration of Helsinki as revised in 2008 under lab protocol PA14-0063."

Please amend your current ethics statement to confirm that your named institutional review board or ethics committee specifically approved this study.

Once you have amended this statement in the Methods section of the manuscript, please add the same text to the “Ethics Statement” field of the submission form (via “Edit Submission”).

"Thermo Fisher Scientific eBioscience provided the QuantiGene Plex Assay for

evaluation purposes. Angle PLC provided the Parsortix® PR1 instrumentation and

GEN3D cell separation cassettes for evaluation purposes."           

"I have read the journal's policy and the authors of this manuscript have the following

competing interests: JMR serves on the scientific advisory board for Angle PLC."

6. We noted in your submission details that a portion of your manuscript may have been presented or published elsewhere.

"Presented in part at the American Association for Clinical Chemistry Annual Conference 2016, Philadelphia PA."

Please clarify whether this conference proceeding was peer-reviewed and formally published. If this work was previously peer-reviewed and published, in the cover letter please provide the reason that this work does not constitute dual publication and should be included in the current manuscript.

7.   We note that Figures in your submission contain may copyrighted images. All PLOS content is published under the Creative Commons Attribution License (CC BY 4.0), which means that the manuscript, images, and Supporting Information files will be freely available online, and any third party is permitted to access, download, copy, distribute, and use these materials in any way, even commercially, with proper attribution. For more information, see our copyright guidelines: http://journals.plos.org/plosone/s/licenses-and-copyright.

a)        You may seek permission from the original copyright holder of Figures to publish the content specifically under the CC BY 4.0 license.

Reviewers' comments:

Reviewer's Responses to Questions

**Comments to the Author**

1. Is the manuscript technically sound, and do the data support the conclusions?

Reviewer #1: Yes

2. Has the statistical analysis been performed appropriately and rigorously? 

Reviewer #1: I Don't Know

3. Have the authors made all data underlying the findings in their manuscript fully available?

Reviewer #1: Yes

4. Is the manuscript presented in an intelligible fashion and written in standard English?

Reviewer #1: Yes

5. Review Comments to the Author

Reviewer #1: Development of an antibody-independent platform for CTCs is highly desirable and this reviewer commends the authors and company.

Since the work is intended to be incorporated into a clinical lab workflow, some comments should be made on the integrity of the PCR assays over time due to amplicon containment requirements.

Other issues from the analytical and operational aspects of the platform itself should be commented on, specifically:

1. deformation and capillary flow required by the platform should be expected to result in lysis of some patient CTCs. How are these phenomena represented in the recovery data? is this the source of spiked cell loss?

2. please expand on the concept of gap size. why is there a decreased recovery with increased gap size?

3. for the hybridization data, I read the manuscript as saying 4 spiked specimens assayed in duplicate for each cell line to generate the data. is that correct?b Can the recovery data for all experiments be pooled to generate a recovery regression model and equation related to spike level?

related to this, figure 1b shows that recovery is cell line dependent. is recovery cell size dependent?

4. In figure 3, are the spiked cell numbers per 5 mL or 1 mL of blood, please state. Also please specify the elapsed time between harvest, spiking and purification.

5. after purification, what is the optimal method of cell preservation prior to qRT PCR analysis?

6. in figure 3C, the SKBR3 10 cell spike recovery level varied over 6 CT values. please comment.

7. This manuscript should include data on performance of this method compared to the cell search on a small panel of human specimens.

6. PLOS authors have the option to publish the peer review history of their article (what does this mean?). If published, this will include your full peer review and any attached files.

Reviewer #1: No

---

## [Author Response · Author response to Decision Letter 0]

20 Aug 2020

(please see "Response to Reviewers" file for formatted version of this text)

Journal Requirements:

1. Style Requirements

a. Will be finalized after peer review if that is acceptable

2. Ethics statement

a. Revised

3. “data not shown”

a. Removed

4. financial disclosure

a. Amended as suggested

5. Competing Interests

a. I have read the journal's policy and the authors of this manuscript have the following competing interests: JMR serves on the scientific advisory board for Angle PLC. This does not alter our adherence to PLOS ONE policies on sharing data and materials.

6. Abstract Publication

a. Data was partially presented as a poster in the American Association for Clinical Chemistry Annual Conference 2016, Philadelphia PA. It was neither peer-reviewed nor formally published. 

7. Figure copyright

a. There are no copyrighted figures in the manuscript, all figures were prepared by the authors. We have added a citation in the Methods section for the Ingenuity Pathway Analysis shown in supplemental figure 4.

Reviewer Comments to the Author:

1. Deformation and capillary flow required by the platform should be expected to result in lysis of some patient CTCs. How are these phenomena represented in the recovery data? is this the source of spiked cell loss?

a. We agree, but quantifying this effect was beyond the scope of this project. We added a statement in the discussion. 

2. Gap Size. Please expand on the concept of gap size. why is there a decreased recovery with increased gap size?

a. Added to discussion, second paragraph. Any population of cells has a distribution of cell sizes, any cells smaller than the gap size will pass through the system without enrichment. A larger portion of this distribution is lost to capture as the gap size threshold is increased.

3. Hybridization Linearity Data: for the hybridization data, I read the manuscript as saying 4 spiked specimens assayed in duplicate for each cell line to generate the data. is that correct? Can the recovery data for all experiments be pooled to generate a recovery regression model and equation related to spike level?

a. Included below is a single regression for KRT18 pooling all data using cell line as a factor, the manuscript shows separate regressions for each cell line. Only technical duplicates are available for the spiked cells, but the reviewer is otherwise correct, the un-spiked samples have 4 biological replicates. As seen in the model below, expression is significantly related to the number of cells spiked (p = 0.000171). We changed the recovery regression R2 values in the manuscript (line 323 tracked version, line 308 clean) to include the replicate points as suggested by the reviewer here, reflecting both linearity and precision. The higher R2 shown before only reflected linearity. 

Recovery regression model

KRT18 Estimate Std. Error t value Pr(>|t|)

(Intercept) 78.0053 422.7937 0.184 0.855

# Cells Spiked 0.4174 0.0958 4.357 0.000171

Cell Line MCF7 control 

Cell Line MDA231 -629 567.602 -1.108 0.277562

Cell Line MDA453 -636.25 567.602 -1.121 0.272183

Cell Line SUM190 946.9375 567.602 1.668 0.106812

Multiple R-squared: 0.521, Adjusted R-squared: 0.45 

Model: lm(KRT18 ~ Cells Spiked + Cell Line)

b. related to this, figure 1b shows that recovery is cell line dependent. is recovery cell size dependent? Yes. Recovery decreases for smaller cells. 

4. In figure 3, are the spiked cell numbers per 5 mL or 1 mL of blood, please state. Also please specify the elapsed time between harvest, spiking and purification.

a. The number of spiked cells is reported per sample of blood enriched by the system, generally at least 5mL. Added to methods and figure 3 legend. 

b. Typically start processing within 4 hours of phlebotomy. Added typical times to methods section.

5. What is the optimal method of cell preservation prior to qRT-PCR analysis?

a. Good question! Immediate lysis is the best solution we have found. We noted that single shot buffers that take the sample directly to PCR without nucleic acid enrichment did not work well, particularly for florescent-based assays (e.g. qRT-PCR) when trace red blood cells remained in the sample. Not added to manuscript.

6. In figure 3C, the SKBR3 10 cell spike recovery level varied over 6 CT values. please comment.

a. The small change in control gene expression suggests this is related to CTC recovery and not a PCR artifact. We felt it was best to include all data without censoring, irrespective of expectations. As the reviewer astutely noted above, enrichment is dependent on cell size, which varies even within a single culture and was not strictly controlled prior to enrichment. Additionally, larger cells contain more RNA. We could also speculate that user-dependent variance in micromanipulation also played a role. Such variances are exacerbated when working with small cell numbers. Not added to manuscript.

7. This manuscript should include data on performance of this method compared to the cell search on a small panel of human specimens.

a. Several publications have compared CellSearch and Parsortix and the citations have been updated in the revised manuscript. Chudziak et al show that enrichment of EPCAM+ cells is similar by CellSearch and Parsortix. PMID: 26605519 DOI: 10.1039/c5an02156a and PMID: 26789903 PMCID: PMC5069649 DOI: 10.1002/ijc.30007

---

## [Decision Letter · Decision Letter 1]

9 Oct 2020

Antigen-agnostic microfluidics-based circulating tumor cell enrichment and downstream molecular characterization

PONE-D-20-19549R1

Dear Dr. Cohen,

We’re pleased to inform you that your manuscript has been judged scientifically suitable for publication and will be formally accepted for publication once it meets all outstanding technical requirements.

Kind regards,

Jeffrey Chalmers, Ph.D.

Academic Editor

PLOS ONE

Additional Editor Comments (optional):

Reviewers' comments:

Reviewer's Responses to Questions

**Comments to the Author**

1. If the authors have adequately addressed your comments raised in a previous round of review and you feel that this manuscript is now acceptable for publication, you may indicate that here to bypass the “Comments to the Author” section, enter your conflict of interest statement in the “Confidential to Editor” section, and submit your "Accept" recommendation.

Reviewer #1: All comments have been addressed

2. Is the manuscript technically sound, and do the data support the conclusions?

Reviewer #1: Yes

3. Has the statistical analysis been performed appropriately and rigorously? 

Reviewer #1: Yes

4. Have the authors made all data underlying the findings in their manuscript fully available?

Reviewer #1: Yes

5. Is the manuscript presented in an intelligible fashion and written in standard English?

Reviewer #1: Yes

6. Review Comments to the Author

Reviewer #1: Thankyou for addressing the points raised in the original review.

Two points to consider going forward:

1. At present, it is not clear exactly what a patient specimen report might look like and what role, if any, quantitation might play in either patient diagnosis or management during therapy. It will be interesting going forward to see if this method overcomes the limitations frequently associated with ctDNA analyais, specifically the fraction of specimens in which there is adequate analyte for statistically significant analysis.

2. Application in monitoring to response to therapy can use at least two different approaches: characterization of changes in CTC phenotypic distribution (EMT status) and how that might be quantified, vs monitoring tumor burden itself as a way to determine therapeutic success. It is not yet clear which approach this method is best suited to.

However, this study lays the groundwork for addressing those important questions.

The major reservation this reviewer sees is determining the minimum cell number in patient specimens that can be analyzed and reported with statistical significance, a problem common to all CTC platforms.

7. PLOS authors have the option to publish the peer review history of their article (what does this mean?). If published, this will include your full peer review and any attached files.

Reviewer #1: **Yes: **Robert J. Kinders

---

## [Editor Report · Acceptance letter]

15 Oct 2020

PONE-D-20-19549R1 

Antigen-agnostic microfluidics-based circulating tumor cell enrichment and downstream molecular characterization 

Dear Dr. Cohen:

I'm pleased to inform you that your manuscript has been deemed suitable for publication in PLOS ONE. Congratulations! Your manuscript is now with our production department. 

Kind regards, 

on behalf of

Dr. Jeffrey Chalmers 

Academic Editor

PLOS ONE